# Recent Advances in Assessing the Clinical Implications of Epstein-Barr Virus Infection and Their Application to the Diagnosis and Treatment of Nasopharyngeal Carcinoma

**DOI:** 10.3390/microorganisms12010014

**Published:** 2023-12-20

**Authors:** Tomokazu Yoshizaki, Satoru Kondo, Hirotomo Dochi, Eiji Kobayashi, Harue Mizokami, Shigetaka Komura, Kazuhira Endo

**Affiliations:** Division of Otolaryngology-Head and Neck Surgery, Graduate School of Medical Science, Kanazawa University, 13-1 Takaramachi, Kanazawa 920-8641, Japan; ksatoru@med.kanazawa-u.ac.jp (S.K.); h_dochi@med.kanazawa-u.ac.jp (H.D.); haruesun@med.kanazawa-u.ac.jp (H.M.); komushige@med.kanazawa-u.ac.jp (S.K.); endok@med.kanazawa-u.ac.jp (K.E.)

**Keywords:** nasopharyngeal carcinoma, Epstein-Barr virus, gene expression, LMP1, BZLF1

## Abstract

Reports about the oncogenic mechanisms underlying nasopharyngeal carcinoma (NPC) have been accumulating since the discovery of Epstein-Barr virus (EBV) in NPC cells. EBV is the primary causative agent of NPC. EBV–host and tumor–immune system interactions underlie the unique representative pathology of NPC, which is an undifferentiated cancer cell with extensive lymphocyte infiltration. Recent advances in the understanding of immune evasion and checkpoints have changed the treatment of NPC in clinical settings. The main EBV genes involved in NPC are LMP1, which is the primary EBV oncogene, and BZLF1, which induces the lytic phase of EBV. These two multifunctional genes affect host cell behavior, including the tumor–immune microenvironment and EBV behavior. Latent infections, elevated concentrations of the anti-EBV antibody and plasma EBV DNA have been used as biomarkers of EBV-associated NPC. The massive infiltration of lymphocytes in the stroma suggests the immunogenic characteristics of NPC as a virus-infected tumor and, at the same time, also indicates the presence of a sophisticated immunosuppressive system within NPC tumors. In fact, immune checkpoint inhibitors have shown promise in improving the prognosis of NPC patients with recurrent and metastatic disease. However, patients with advanced NPC still require invasive treatments. Therefore, there is a pressing need to develop an effective screening system for early-stage detection of NPC in patients. Various modalities, such as nasopharyngeal cytology, cell-free DNA methylation, and deep learning-assisted nasopharyngeal endoscopy for screening and diagnosis, have been introduced. Each modality has its advantages and disadvantages. A reciprocal combination of these modalities will improve screening and early diagnosis of NPC.

## 1. Introduction

Human gamma herpesviruses, including the Epstein-Barr virus (EBV; human herpesvirus 4; HHV-4) and Kaposi’s sarcoma-associated herpesvirus (KSHV; human herpesvirus 8; HHV-8), have oncogenic properties. EBV and KSHV are lymphotropic viruses, but they also infect epithelial cells, and their infected cells may develop carcinomas [1]. EBV was the first human oncovirus discovered in Burkitt’s lymphoma (BL) in 1964 [2]. Kenyan patients with nasopharyngeal carcinoma (NPC) whose sera were selected as controls in a BL study showed more antibody precipitation than those with BL [3]. Four years later, EBV was identified in undifferentiated NPC cells [4]. The definite association of EBV with NPC was confirmed by the ubiquitous identification of EBV DNA in the tumor cells of NPC tissues in endemic and non-endemic areas for NPC [5]. Based on outstanding papers published in the past half-century, we introduce new insights into the relevance of the unique clinicopathological characteristics of NPC associated with EBV infection.

### 1.1. Histopathology of NPC and EBV Infection

EBV infection is closely associated with pathological characteristics of NPC. The World Health Organization (WHO) classification system, which is based on the grade of differentiation of tumors, is generally accepted for the pathological classification of NPC. WHO grades I, II, and III represent keratinizing, non-keratinizing-differentiated, and non-keratinizing-undifferentiated NPC, respectively. WHO II and III are considered to indicate EBV-associated NPC [6]. However, there has been a debate on the relevance of EBV in WHO I NPC diagnosis. A previous prevalent opinion was that WHO I NPC was originally an EBV-driven tumor. Throughout the tumor progression, EBV escaped or was eliminated from tumor cells. Thus, a small amount of EBV DNA was detected in WHO type I tumors. However, a recent prevalent opinion based on serological and histological studies is that WHO I NPC is a squamous cell carcinoma similar to general head and neck carcinoma.

EBV-associated WHO II and III NPCs are characterized by the prominent infiltration of lymphocytes in the tumor-surrounding area and the so-called lymphoepithelioma. This difference in histological features has been investigated in various directions regarding the association between EBV and the clinical features of NPC.

WHO I NPC accounts for less than 20% of NPC cases globally. This rate is relatively low in endemic areas, such as Southeast Asia and southern China. In other words, the prevalence of non-keratinizing (WHO II, III) NPC is higher in endemic areas (>95%) and is predominantly associated with EBV infection. However, patients with NPC present elevated IgG and IgA concentrations in response to the viral capsid antigen (VCA) and early antigen (EA) of EBV regardless of endemic or non-endemic area [7]. For WHO I NPC, initial studies reported similar pathology and EBV serologic profiles similar to those of other head and neck carcinomas [8,9], whereas other studies have suggested that all types of NPC result in elevated concentrations of EBV antigens [10]. Recent high-resolution analyses, such as duplex multiplex assays for EBV IgA and IgG antibodies, have shown that very fine adjustment of sample sera is mandatory for the precise quantification of antibody titers [11,12]. Presumably, the mixed review of the serological association of WHO I NPC is attributable to technical problems. However, it is generally accepted that WHO I represents NPC that is unrelated to EBV, and WHO II and III represent EBV-associated NPC. The clinical characteristics of WHO I NPC differ from those of WHO II and III NPCs. A multicenter prospective trial revealed that patients with WHO I NPC had no distant metastatic recurrence, and all relapsed sites were locoregional areas. This pattern is similar to the patterns of conventional head and neck cancers. In contrast, patients with WHO II and III NPCs had significantly higher rates of distant metastatic recurrence. These results indicate that EBV contributes to the high metastatic properties of WHO II and III NPCs [13] (Table 1).

Recent trends of human papillomavirus (HPV) infections in head and neck carcinomas, especially oropharyngeal carcinomas, have been incorporated into NPC research. HPV was detected in two of the 58 NPC tissues at our institute. The HPV-positive cases were classified as WHO II, and EBERs were detected using in situ hybridization. One case had HPV18, and the others had HPV16 and 18 [14].

Reports have been inconsistent, likely because of the limited number of patients and the ethnic and geographic differences among the study populations. Several studies have detected HPV in NPC, with some demonstrating a dichotomy between EBV and HPV infections predominantly in non-endemic regions [15,16,17]. Others have reported cases of EBV and HPV co-infection, predominantly in patients from endemic regions [18,19,20]. Similar to EBV, a serological test has been developed for HPV and applied to screening for HPV-associated diseases [21].

### 1.2. EBV Gene Expression in NPC

The default program for EBV infection in nasopharyngeal epithelial cells is lytic infection. Therefore, switching the mode of infection from lytic to latent and maintaining latent infection are essential steps in NPC pathogenesis. EBV latency programs can be categorized into three types. Latency affects various diseases and infected cell types [22,23,24]. Two EBV-encoded genes, EBER and EBNA1, are expressed in all three types of latencies. Type I latency is characteristic of Burkitt’s lymphoma, and it is defined as the expression of a minimal number of latent EBV genes, which are EBER and EBNA1. For type II latency, LMP1 and LMP2A are expressed in addition to EBER and EBNA1. Type II latency is observed in NPC, Hodgkin’s disease, and T/NK-cell lymphoma. Type III latency is recognized in lymphoproliferative diseases in immunocompromised hosts such as organ recipients or HIV patients [25,26]. The remarkable expression of *BART-*miRNAs in NPC and EBV-associated gastric cancer strongly suggests an important carcinogenic role for *BART-*miRNAs in EBV-associated epithelial malignancies [27].

NPC is an EBV type II latency malignant tumor that expresses three EBV-encoded proteins (EBNA1, LMP1, and LMP2) and two EBV-encoded transcripts, EBERs and BARTs [22,23,25]. Among the EBV genes, we have mainly focused on the EBV primary oncogene LMP1 as it promotes all aspects of “the hallmark of cancer” advocated by Hanahan and Weinberg [28,29].

### 1.3. Effect of LMP1 Expression in NPC

LMP1 expression is generally observed in moderate-to-severe dysplastic lesions in the EBV-infected nasopharyngeal epithelium or preinvasive NPC. Thus, LMP1 plays an important role in driving EBV-infected premalignant cells into the early stages of NPC [30]. The comprehensive study on LMP1 revealed its oncogenic properties by activating multiple cellular signal pathways such as NF-κB, AP-1, and PI3 kinase [31]. Among them, the activation of the NF-κB pathway is a major characteristic of LMP1. Several steps of carcinogenesis, including anti-apoptosis, anti-cellular stresses, cell metabolism, acquisition of cancer stemness, and immune attacks, are mediated by the NF-κB-associated pathway [32,33,34,35,36,37,38,39]. Moreover, the contribution of LMP1 to the highly metastatic features of NPC, represented by the downregulation of cell adhesion, upregulation of stromal destruction, and angiogenesis, is also attributable to this NF-κB signaling [40,41,42,43,44] (Figure 1).

Interestingly, the mutually exclusive relationship of LMP1 expression with somatic mutations of negative regulators that are located upstream of the NF-κB pathway in NPC has been reported from the endemic region. These findings indicate a special role of NF-κB activation via constitutive activation through somatic mutations dominated by CYLD, TRAF3, NFKBI, and NLRC5, or LMP1 in the pathogenesis of NPC [45,46]. These somatic alterations of NF-κB regulators may supplant the need for LMP1-mediated NF-κB activation in tumor cells during tumor progression [46].

## 2. Effect of BZLF1 Expression in NPC

Two EBV infection modes, the latent and lytic phases, are believed to be tightly regulated by specific gene expression patterns; however, evidence of the simultaneous expression of latent and lytic genes within the same cell has been accumulated. Among lytic EBV gene products, we focus on BZLF1 as it is the most studied and well-known initiator of the EBV replication switch.

BZLF1, identified in studies on the defective EBV genome, is a trigger gene that facilitates the progression of EBV infection from the latent to the lytic phase [47,48]. The induction of BZLF1 in latently infected cells usually activates the EBV lytic cascade. However, this lytic cascade terminates without producing mature EBV virions; this is called the abortive lytic infection [49]. Thus, the BZLF1 protein ZEBRA can affect host cells during abortive lytic infection [49].

The roles of ZEBRA in EBV infection, the lytic cycle, and oncogenesis have been extensively studied. ZEBRA is an AP-1 mimicking transcriptional factor. Thus, ZEBRA transactivates EBV genes, as well as various host cellular genes related to proliferation, inflammation, angiogenesis, and metastasis. ZEBRA downregulates apoptosis-related genes, MHC class II genes, and interferon regulatory factors, eventually acquiring resistance to cell death and evasion during immune surveillance [38]. ZEBRA also interacts with various cellular proteins, including NF-κB, and changes their functions [50,51,52]. Thus, the effects of ZEBRA are complex. The fate of EBV-infected cells during abortive lytic infection requires further investigation.

Both LMP1 and ZEBRA expression have been identified in NPC tumor tissues [48,52]. However, ZEBRA is still clinically valuable. The expression of the protein in NPC tissues and elevation of anti-ZEBRA IgG concentrations in sera are biomarkers for poor prognosis in patients with NPC, as well as the diagnosis of early-stage NPC [50,53,54,55]. In addition, elevated concentrations of IgA antibodies against the EBV VCA are high-risk markers for NPC development [56]. Studies on EBV strains isolated from NPC and other EBV-associated diseases suggest the existence of aberrant strains predisposed to spontaneous lytic replication and specific EBV-associated diseases [57,58,59,60,61].

### 2.1. Genetic and Epigenetic Alteration in NPC Host Genome

LMP1 and BZLF1 are key EBV genes that drive EBV-infected epithelial cells toward NPC. However, the expression of EBV genes alone is insufficient for the development of clinical NPC. The accumulation of multiple irreversible somatic genetic alterations, in addition to EBV infection, is required for the development of NPC [22]. To date, there have been several reports of frequently identified genetic changes in patients with EBV-associated NPC. The frequent identification of allelic loss at chromosomal loci 3p21.3 and 9p21 in both EBV-positive and -negative pre-invasive lesions suggests that these genetic changes have occurred before EBV infection [22,62,63]. These genetic alterations imply inactivation of tumor suppressor genes. Among the genes, RAS association domain family 1A (RASSF1A) and cyclin-dependent kinase inhibitor 2A (p16/CDKN2A) are candidate genes involved in the initiation of NPC [22,64], as inactivation of RASSF1A and p16 may provide a growth advantage for the clonal expansion of EBV-infected premalignant epithelial cells.

Recent genome-wide analyses of invasive and pre-invasive NPC have elucidated common somatic alterations in the NPC host genome. Focal loss of alleles on chromosomes 3p, 9p, 11q, 13q, 14q, and 16q and amplification of chromosome 11q13, where the cyclin D1 loci are located, are frequently observed [64,65,66]. Among these genes, the inactivation of p16/CDKN2A by homozygous deletion and promoter hypermethylation has been consistently found in almost all NPC samples examined [67], suggesting that the expression of the p16 gene product is commonly downregulated in NPC.

Epigenetic changes can be induced by EBV infection, and they eventually play a role in tumorigenesis [68]. CpG hypermethylation is a common mechanism involved in the inactivation of tumor suppressor genes in human cancers and has been observed in EBV-associated malignant tumors, including Burkitt’s lymphoma, NPC, and EBV-associated gastric cancer [69,70]. A specific EBV epigenetic signature has been more thoroughly studied in EBV-associated gastric cancer, whereas that in EBV-associated NPC is slightly less defined. Nonetheless, promoter methylation of multiple genes at chromosomes 3p21.3 (including RASSF1A [71]), 6p22.1-21.3 (including the MHC locus [72]), and 9p21 (including CDKN2A (p16), CDKN2B (p15) [73], and others [74]) has been identified. To date, the relationship between latent EBV infection and the methylation profile observed in NPC has not been fully elucidated.

ZEBRA acts as a pioneer factor. Considering that ZEBRA is a homologue of c-Fos and binds to the AP-1 site, it is reasonable to speculate that ZEBRA expression in NPC tumor cells could modulate gene expression in both heterochromatin and euchromatin regions. Further studies are needed to explore the broad effects of ZEBRA expression in NPC.

### 2.2. Potential Risk Factors Other Than EBV

In addition to host genetics and EBV infection, other potential risk factors identified by epidemiological studies include the family history of nasopharyngeal carcinoma, active and passive tobacco smoking, consumption of preserved foods and alcohol, and oral hygiene [75,76,77,78,79]. In addition to these traditional carcinogenic factors, the role of microbiota in the pathogenesis of NPC has been gradually elucidated.

Microorganisms within the gut and other niches may contribute to carcinogenesis, influence cancer immunosurveillance, and respond to immunotherapy. Thus, targeting the gut and tumor microbiota in cancer is an issue of interest [78]. The association of EBV has been firmly established as a microbial risk factor for NPC. However, the crucial role of the nasopharynx as a niche of the upper respiratory tract microbiome and the clinical implications of the microbiota in NPC tissues remain largely obscure.

Qiao et al. conducted a transcription study on 12 paired NPC tissues with either high or low bacterial loads. Proliferative genes in cell cycle pathways and metastasis-associated pathways were upregulated in tumors with a high bacterial load. Conversely, both T- and B-cell-mediated immune responses and interferon pathways, as well as O-linked glycosylation, are activated in tumors with a low bacterial load [80]. Subsequently, they conducted a multicenter cohort study that included 802 patients with NPC. They confirmed the existence of microbiota mainly originating from the nasopharynx within NPC tissues. Moreover, the intratumoral bacterial load was negatively associated with tumor-infiltrating immune cells and positively associated with poor prognosis in patients with NPC [81]. The results also showed that the levels of expression of various immune-related genes, such as *CXCL13*, were negatively associated with the load of intratumoral bacteria, particularly *Porphyromonas.* Butyrate produced by *Porphyromonas gingivalis* contributes to the regulation of histone acetylation, which reactivates EBV, implying the existence of crosstalk between the bacteria and EBV [82].

### 2.3. Prognostic Relevance of Cell-Free EBV DNA

The UICC/AJCC staging system has been evolving, and the eighth system is of clinical use [83]. However, this diagnostic system is anatomically based and has limitations in predicting the prognosis or treatment benefits because it is affected by other clinical factors and molecular biomarkers. Several researchers have attempted to incorporate these factors into the system to establish better predictive markers of clinical outcomes.

Among the candidate factors, the combination of pretreatment plasma EBV DNA and clinicopathological variables is a reliable system for arriving at a more accurate prognosis for patients with NPC [84]. Other biomarkers, such as somatic and EBV DNA methylation and miRNA and mRNA expression, have also shown prognostic value and potential clinical applications in patients with NPC [85,86]. Genome-wide screening techniques have revealed the prognostic value of gene expression-based signature analysis and may be beneficial in selecting patients for more intensive treatments [85,86]. However, the diagnostic and prognostic value of circulating EBV DNA is commonly accepted, and quantification kits for circulating EBV, including serum, plasma, and whole blood samples, are now commercially available. These findings have promoted the use of circulating EBV DNA in NPC treatment.

The successful incorporation of pretreatment plasma EBV DNA into the eighth edition has been reported [87,88]. Furthermore, both pretreatment and EBV DNA responses after induction chemotherapy or post-treatment are useful predictors of clinical outcomes [89,90]. These observations have suggested that EBV DNA is a promising liquid biopsy-based biomarker for risk-stratified treatment adaptation.

Longitudinal EBV DNA quantification during and after NPC treatment is now accepted as a valuable method of monitoring NPC. Despite the well-established value of EBV DNA in NPC monitoring, international standardization of assay methods, such as variations in the number of target repeats in the EBV genome, has been a huge obstacle for prospective large-scale trials, even when applying the same assay system [91,92]. In 2016, at the National Cancer Institute EBV testing harmonization workshop for nasopharyngeal carcinoma, NPC and laboratory medicine experts presented the limitations of current assays for EBV DNA quantification and discussed methods for improving combination assays in the future [93]. Studies on the application of EBV DNA as a biomarker for NPC staging are ongoing.

## 3. Immune Microenvironment of NPC and EBV

As mentioned above, EBV-associated NPC is characterized by the prominent infiltration of lymphocytes in the tumor-surrounding area and the so-called lymphoepithelioma. Based on the unique pathological features of NPC, the relationship between the immune system and NPC tumor cells has been intensively investigated.

The nasopharynx contains a unique lymphoid tissue called the nasopharynx-associated lymphoid tissue (NALT), which differs from other lymphoid tissues involved in organogenesis. NALT plays an important role in the mucosal immune system [94]. The secretion of anti-EBV IgA from the NALT into the airway lumen and serum is a marker for NPC screening and diagnosis. New insights into the tumor microenvironment of NPC have been growing, including the tumor–immune microenvironment (TIME). The tumor microenvironment is mainly composed of heterogeneous cellular components such as tumor cells, fibroblasts, endothelial cells, and leukocytes. It also contains various mediators, such as cytokines and exosomes, which influence the behavior of NPC tumors and, eventually, the prognosis of NPC patients. The TIME consists of immunostimulant and immunosuppressive components. Thus, the TIME component also serves as a prognostic biomarker and a potential target for novel therapies [95].

Tumor-infiltrating lymphocytes (TIL) in NPC tissue contain various immune cells, and the proportion of the immune population dynamically changes with tumor progression and treatment. However, TIL levels usually reflect the treatment outcomes of patients. Abundant intratumoral and stromal TILs are predictive markers of favorable outcomes in patients with NPC [96].

### 3.1. Immune Evasion Mechanism in NPC

Malignant tumor cells begin the construction of the TIME once they are recognized as targets of immune cells. Subsequently, the tumor cells escape the immune attack. Generally, viral proteins are presented as antigens combined with MHC class I molecules to recruit CD 8-positive cytotoxic T cells (CTL) to the virus-infected tumor cells. Three EBV genes, EBNA1, LMP1, and LMP2, are expressed in NPC. However, EBV-encoded gene products expressed in NPC cells must escape from immune cells, mainly from CTL, for the development of NPC.

In the EBV latent infection program, the most efficient strategy for immune escape is to keep the level of expression of the EBV antigen per cell as low as possible. Maintaining low antigen expression leads to low CTL epitope presentation in EBV-infected cells. Crotzer et al. reported that the presentation of PRIYDLIEL-like epitopes in EBNA3C is less than one epitope per cell [97], resulting in very low or no detection of EBV-transformed B-cells by CTL clones with high affinity to the peptide [98].

In addition to the low levels of latent gene expression, EBNA1 contains a Gly-Ara repeat domain that suppresses the translation of EBNA1 and prevents the processing of EBNA1, resulting in the inhibition of antigen presentation by MHC class I [99].

Upon activation, LMP1 localizes and aggregates within lipid rafts on the cell membrane [100]. LMP1 mutants that lose their aggregating properties at lipid rafts are recognized by HLA epitope-sensitized CTL, whereas wild-type and LMP mutants that retain their aggregating properties at lipid rafts are not. These results suggest that the aggregation of LMP1 in lipid rafts protects LMP1 from being processed and presented with MHC class I, which allows LMP1 expression in NPC cells to escape immune surveillance [101].

In contrast, lytic infection involves the expression of more than 60 viral gene products with high copy numbers per cell [102]. For the transmission of EBV from carriers who have established T-cell immunity to the EBV gene to other individuals, cells with EBV lytic infection must escape T-cell recognition long enough to produce mature EBV particles. Even in cells under abortive lytic infection, the expression of immunogenic ZEBRA should affect the tumor–immune cell interactions.

BZLF1 suppresses the transcription of inflammatory factors TNF-α and IFN-γ and prevents their response during the EBV lytic infection. This implies that the EBV lytic cycle employs a distinct strategy to evade the antiviral inflammatory response [103].

### 3.2. Superantigen Induction by EBV

One theory states that despite these mechanisms of immune evasion, some EBV gene expression leads to lymphocyte infiltration. This is one theory. The other theory is the induction of superantigens by EBV genes. Superantigens are microbial pathogen-derived proteins that induce strong T-cell responses. HERV-K18 was the first human endogenous provirus to originate from a retrovirus with superantigen activity [104]. HERV-K18 preferentially activates human TCRBV13 T cells [105]. Essentially, it exists in a dormant state. However, the expression of its *env* gene was observed in latently infected EBV cells. Sutkowski showed that both LMP1 and LMP2A are able to transactivate the HERV-K18 superantigen. Both LMP1 and LMP2A are expressed in NPC cells [106]. HERV-K18 may induce heavy T-cell infiltration, and the secretion of cytokines by the interaction of superantigen-expressing tumor cells with activated T-cells can contribute to the highly metastatic features of NPC [107,108].

### 3.3. Modulation of Immune Checkpoint in NPC

Generally, the number of intratumoral and stromal TILs is associated with a favorable prognosis in patients with NPC [96]. The programmed death-1/programmed death-ligand 1 (PD-1/PD-L1) axis plays an important role in T-cell tolerance and immune escape of tumor cells. PD-L1 expression is higher in EBV-positive than in EBV-negative NPC cell lines. It can be due to the abundant IFN-γ induction by EBV infection. In addition, LMP1 and IFN-γ pathways cooperate to regulate PD-L1 [44]. LMP1 knockdown suppresses PD-L1 expression in EBV-positive cell lines. IFN-γ upregulates PD-L1 expression independently and synergistically with LMP1 in NPC tissue [109].

PD-1 is expressed on TIL, while PD-L1 is expressed in tumor and stromal cells. The interaction between PD-1 and PD-L1 suppresses CTL attacks on tumor cells [109]. Thus, the co-expression of PD-1 and PD-L1 in the tumor microenvironment usually predicts the recurrence and metastasis of NPC after initial therapy. Antibodies that block this pathway have been clinically used against various cancers, including NPC [95,110]. Adoptive T-cell therapy targeting the EBV gene products is a potential therapeutic strategy for NPC. PD-L1 expression in cancer cells can inhibit the effector functions of adoptively transferred EBV-specific T-cells. The combination of EBV-specific adoptive T-cell and PD-L1 blockade therapies has been reported to be more effective [111].

Several immunotherapies have been explored for NPC; however, these trials have yielded inconsistent conclusions, probably due to different immune microenvironments [109,110,112,113,114,115]. For example, LMP1 promotes myeloid-derived suppressor cell (MDSC) expansion in the tumor microenvironment by promoting extra-mitochondrial glycolysis in malignant cells. In addition to RAGE, LMP1 promotes the expression of multiple glycolytic genes, such as GLUT1. This metabolic reprogramming induces the NOD-like receptor family protein 3 inflammasome and activates the arachidonic cascade, which in turn activates various cytokines such as IL-1β, IL-6, and GM-CSF. These changes in the tumor environment result in NPC-derived MDSC induction [116].

### 3.4. Challenge for EBV Vaccine Development

The close association of EBV infection with NPC suggests that controlling EBV infection and targeting EBV-infected cells are effective strategies for preventing NPC development and managing developed NPC. However, despite the urgent need for prophylactic or therapeutic solutions, there has been no licensed EBV vaccine thus far [117]. The major obstacles to developing a prophylactic vaccine include the global prevalence of EBV and the fact that infection typically occurs during early childhood to adolescence, resulting in a limited number of control cohorts without EBV infection. In addition, the complexity of the EBV replication system and its infectivity to T lymphocytes and NK cells present challenges in eliminating all EBV target cells [117]. With recent advancements in messenger RNA vaccines, exemplified by their successful application in the COVID-19 pandemic, various viruses, including EBV, have been considered candidates for this technique [118]. Moderna has announced the initiation of a phase 1 study for its mRNA Epstein-Barr virus (EBV) vaccine, code-named mRNA-1189. mRNA-1189 comprises five mRNAs encoding envelope glycoproteins (gp320, gH, gL, gp42, and gp220) that bind to target cell surface receptors, playing key roles in initiating EBV infection of target cells. The administration of mRNA-1189 aims to induce a broad immune response that could prevent EBV infection in various types of cells, ultimately reducing the symptoms of infectious mononucleosis. According to the manufacturer, preclinical testing of mRNA-1189 in mice and nonhuman primates demonstrated high and durable levels of antigen-specific antibodies against B cell and epithelial cell infection [119]. The announced phase 1 clinical trial (NCT05164094) will assess the safety and tolerability of three different doses of mRNA-1189 in healthy adults aged 18 to 30. The humoral immune response will be evaluated up to day 197 [120]. (https://clinicaltrials.gov/study/NCT05164094 (accessed on 11 October 2023)).

mRNA-1189, which encodes additional latent EBV genes, may serve as a candidate for therapeutic vaccines. Certainly, mRNA technology is poised to bring about significant progress in the future of vaccinology.

## 4. Early Diagnosis of NPC

The prognosis of patients with advanced-stage disease, including NPC, has not improved despite various treatment modalities and the systematic clinical trials that have been developed. Currently, patients with early-stage NPC are treated with radiotherapy alone, whereas those with an advanced stage are treated with a combination of chemotherapy and radiotherapy. Patients with advanced-stage NPC have worse survival rates and quality of life than those with early-stage disease [9,117]. Thus, early detection and timely treatment of NPC is an effective strategy to improve prognosis and quality of life and reduce the medical burden of patients with NPC [118,119]. Nasopharyngeal endoscopy, EBV serology, and plasma EBV DNA have been introduced and assessed by several researchers for the early detection of NPC. Here, we discuss the advantages and disadvantages of these modalities (Table 2).

Nasopharyngeal endoscopy with pathological examination of suspected lesions is the gold standard for the diagnosis of NPC [120]. However, it is primarily used for individuals with suspected NPC. Therefore, it is not suitable for early screening, especially large-scale screening. In addition, current laboratory blood tests (e.g., anti-EBV antibody screening and EBV DNA testing) can provide high specificity and sensitivity, but they still have low positive predictive values (PPVs) and produce more false-positive results, leading to repeated nasal endoscopic examinations, biopsies, and follow-up. To improve the early diagnosis rate of NPC, He et al. developed a deep-learning-based NPC detection model based on a large-scale video frame dataset for nasopharyngeal endoscopy, which enabled more accurate biopsy site selection during endoscopy [121].

There are currently three common clinical EBV antibody detection targets: VCA-IgA, EBNA1-IgA, and EA-IgA [122]. The responses to these antibodies are markedly heterogeneous, and the response to a single antibody is not sufficient for NPC screening. Moreover, the traditional immunoenzyme labeling method is complicated, and sometimes the results are reproducible. Therefore, it is gradually being replaced by ELISA. With the development of new techniques for the detection of dual and triple antibodies, the sensitivity and specificity of the serological diagnosis of NPC have significantly improved [123]. In recent years, researchers have made efforts to improve the sensitivity, specificity, and PPV for detecting early-stage NPC. However, an acceptable PPV (<10%) has not been achieved.

The IgA serological test is the most generally accepted NPC screening method, but the quantification of plasma EBV DNA originating from the tumor is a more sensitive biomarker for screening, as well as for predicting and detecting recurrent NPC disease. Compared with serological screening, plasma EBV DNA detection is more sensitive and specific. The PPV of plasma EBV DNA is also better than that of antibody testing but still lower than 20%. In addition, the cost of EBV DNA testing is high. EBV DNA testing requires advanced equipment and standardization among laboratories that use this method, making it more difficult to scale out EBV DNA testing in populations from all high-risk areas [123].

In a recent review by Yuan et al., EBV detection in nasopharyngeal swabs as an additional test for EBV serology in high-risk individuals reduced the number of candidates for close follow-up. This method achieved a 40% improvement relative to serological examination alone [124]. However, the PPV remained low. The hypermethylation status of gene promoters in nasopharyngeal swabs can also be used for NPC screening. However, the sample sizes of the relevant studies were not sufficiently large. Large-scale studies are necessary to assess its effectiveness for NPC screening [125,126] (Table 2).

The anti-BNLF2b total antibody (P85-Ab), a novel biomarker, was validated through a large-scale prospective screening program and compared with the two standard antibodies, EBNA1-IgA and VCA-IgA.

Forty-seven patients with NPC (38 with early-stage disease) were identified from a cohort of 24,852 prospectively eligible participants. P85-Ab showed higher sensitivity (97.9% vs. 72.3%; ratio, 1.4 [95% CI, 1.1 to 1.6]), specificity (98.3% vs. 97.0%; ratio, 1.01 [95% CI, 1.01 to 1.02]), and PPV (10.0% vs. 4.3%; ratio, 2.3 [95% CI, 1.8 to 2.8]) than the combination of EBNA1-IgA and VCA-IgA. The combination of anti-BNLF2b total antibody with anti-EBNA1-IgA and anti-VCA-IgA improved the PPV to 44.6% (95% CI, 33.8 to 55.9), with a sensitivity of 70.2% (95% CI, 56.0 to 81.4). Serological screening using these three antibodies may provide a promising novel biomarker [127].

## 5. Conclusions

Clinicopathological features of NPC are mostly attributable to EBV infection and, eventually, virus–host and tumor–immune interaction. The infection program of EBV in NPC is basically a type II latent infection; however, EBV genes involved in the lytic infection program are also expressed. Among both program genes, latent gene LMP1 and lytic gene BZLF1 are the main players in organizing the clinicopathological characteristics of NPC. The elevation of EBV antibodies and cell-free DNA in both patients and future candidates for NPC represents crucial features. These features provide valuable insights into the development of more sensitive and cost-effective screening and diagnostic methods. The unique TME characterized by an intense infiltration of immune cells also offers clues for potential therapeutic targets in NPC. In fact, specific inhibitors aimed at immune evasion mechanisms, such as immune checkpoint inhibitors, are now in clinical use. Early detection and the development of novel drugs for advanced NPC constitute effective strategies to enhance the prognosis of patients with NPC. Efforts to clarify the mechanism of EBV-mediated NPC pathogenesis and apply the findings to NPC treatment are ongoing. With the rapid technological advances, further evidence is expected to be revealed.

## Figures and Tables

**Figure 1 microorganisms-12-00014-f001:**
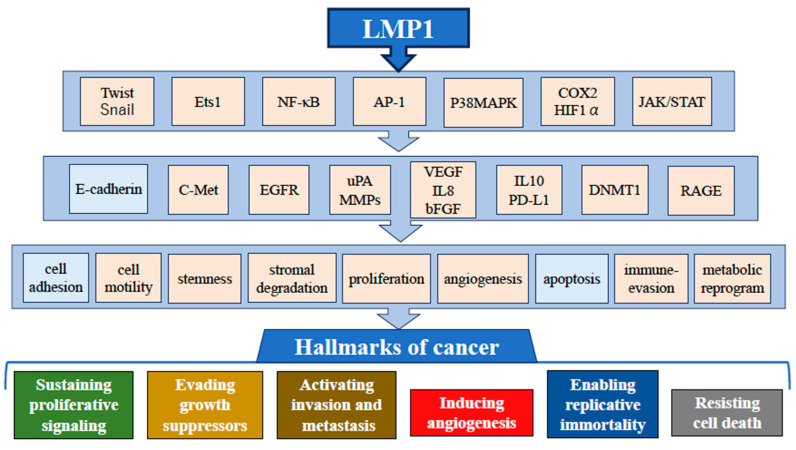
Activation of multiple cell signaling pathways by LMP1. LMP1-mediated signal activation promotes cell survival, proliferation, angiogenesis, invasion, metastasis, metabolic reprogramming, and cytokine production. LMP1 downregulates E-cadherin expression, which ultimately affects cell adhesion. As a result, LMP1 promotes six generations of “hallmarks of cancer.” Factors downregulated by LMP1 are indicated by light blue-colored boxes.

**Table 1 microorganisms-12-00014-t001:** Nasopharyngeal cancer histology and clinicopathological features.

	WHO I	WHO II	WHO III
Differentiation status	well differentiated	moderately to poorly differentiated	undifferentiated
Histological category in WHO classification	keratinizing	nonkeratinizing-differentiated	nonkeratinizing-undifferentiated
TIL infiltration	fair to moderate	heavy	
EBERs in tumor	(−) or faint	(+)	
EBV antibodies	not elevated	elevated	
Chemoradiosensitivity	moderate	good	
Metastatic property	low to moderate	high	
Epidemiology	20% in non-endemic area; <5% in endemic areas	80% in non-endemic areas; >95% in endemic areas	

EBERs; EBV-encoded small RNAs, TIL; tumor-infiltrating lymphocytes, WHO; World Health Organization.

**Table 2 microorganisms-12-00014-t002:** Comparison of modalities for NPC screening and early diagnosis.

Screening Procedure	Swab Cytology	Serological Test	Circulating EBV-DNA	Cell Free DNA Methylation	Nasopharyngeal Endoscopy
Benefits	Easy sampling	Most popular methods	Higher sensitivity and specificity	Higher sensitivity and specificity	Higher positive predictive value of approximately 90%
	Quick detection	Possible combination of multiple antibody types	Higher positive predictive value but still <20%	Higher positive predictive value of approximately 90%	Combination of deep learning model improve accuracy
				Applicable for EBV negative NPC	Applicable for EBV negative NPC
Drawbacks	Influenced by sample quality, physician’s skill, cytologist’s skill	Low level of positive predictive value (10%)	Influenced by laboratory skill	More advanced technique required	Large scale screening is not possible
			Not globally standardized	Lack of large-scale study	Higher cost
			Higher cost

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
