# Peer review of "Recent Advances in Assessing the Clinical Implications of Epstein-Barr Virus Infection and Their Application to the Diagnosis and Treatment of Nasopharyngeal Carcinoma"

_microorganisms, 2023, doi:10.3390/microorganisms12010014_

Round 1
Reviewer 1 Report
Comments and Suggestions for Authors
This review article has the potential to be a timely addition to the field, however, in general it is too brief to provide sufficient added value to existing knowledge. I would recommend revisiting the core message and then redrafting certain sections.
Firstly, the title does not clearly articulate the content to the reader - what does "clinicopathological relevance of Epstein-Barr virus infections" mean? It has long been known that EBV is associated with NPC. What does this review add to the body of knowledge? What is unique or novel about this? Integrate THIS into your title.
Similarly with the abstract, this should draw out the key points addressed in the review, but it fails to do so clearly. I think it just needs to be more clearly stated.
The subsections on LMP1 and BZLF1 are very short - too brief really. Please expand to include more recent, relevant references too.
Then, there are numerous issues throughout, which are laid out line by line below:
In addition to the comments from the English language editor throughout:
Lines 21-23: This is not a complete sentence. (Or perhaps you mean “through” instead of “though”?)
Line 32: Kaposi’s (not Kaposi)
Line 53: Formatting – please indent for consistency
Line 53: Rephrase as “EBV-associated WHOO II and III NPC are characterized…”
Line 66: Please add a reference at the end of this sentence (“…WHO II and III NPCs.”) – unless it is the same reference as at the end of line 69, in which case you can ignore this comment.
Table 1: As the English language editor says, the Table needs a footnote with a list of abbreviations in alphabetical order. Please see author guidelines for more detail.
Line 75: Please consider a different word than “adapted” as this sentence does not make sense.
Line 75: The following sentence is also inaccurate, “HPV is considered to contribute to WHO grad I NPC because EBV genomes are frequently missing from samples of this subtype.” The two facts are unrelated – you cannot possibly consider HPV causality merely because EBV genomes are missing. Please rephrase for clarity.
Line 83: This sentence lends nothing to the argument. Either expand it to give it more weight and meaning (e.g., WHY is NPC of interest?) or remove it.
Line 88: Remove “the” before “latent”
Line 88: Remove “The” before “EBV”
Line 97: Your statement that any of the latent gene products influence tumorigenicity, but this is not quite true. EBNA1 does not promote tumorigenicity but rather episomal maintenance. Please be very clear and accurate in your writing.
Line 110: Please rephrase the end of this sentence – which signaling? Presumably signaling by LMP1, but this needs to be made clearer.
Line 114: The “k” in NF-kB is not in symbol format, whereas it is elsewhere. Please go through the entire document and ensure consistency throughout.
Lines 114-115: This sentence is rather “lost” here on its own. There needs to be more context around this statement. What kind of somatic alterations? How does LMP1 overexpression drive cytotoxicity?
Section 4 & 5: Given the importance of LMP1 and BZLF1 on NPC pathogenesis, you have written very little about either gene product. I would recommend expanding these sections to be more comprehensive.
Line 121: This is not 100% accurate – any of the lytic gene products influence NPC tumorigenicity – I would temper this statement. You made a similar claim earlier about latent gene products too (line 97), which also wasn’t accurate.
Line 142: How can LMP1 and BZLF1 genes precede EBV infection in NPC? The virus is what encodes them, therefore the infection surely precedes their expression! Please rephrase
Lines 142-143: In what way was the influence of these genes reversible? Do you have a reference for this?
Line 203: Formatting – please indent for consistency
Line 231: Please reconsider use of the word “should” – this sentence could be more concise (e.g., “Malignant tumor cells begin constructing the TIME once they are recognized as immune cell targets.”)
Line 243: Repetition of derivatives of the word “aggregate” – consider rephrasing
Line 258: What is meant by “leaky EBV gene expression”?
Lines 258-259: Please incorporate “This is one theory” into the preceding sentence (e.g., “One theory states that…”)
Line 262: Please add “gene” after “env”
Line 262: Please remove “the” before “latently”
Line 263: Why separate out LMP1 and LMP2A if they both transactivate the same gene? If this is in fact correct, please combine (e.g., “both LMP1 and LMP2A are able to transactivate…”
Line 264: LMP1 and LMP2A are expressed in NPC cells – please specify which subtype
Lines 271-272: Please add appropriate references for these statements, unless it is incorporated within ref 110, in which case ignore this comment
Line 290: Please find an alternative word to “acceptable” as this does not make sense
Line 313: Please remove “as mentioned in the previous chapter” as this is not relevant here (I’m assuming this came from a student thesis?!)
Table 2: In addition to the English language editor’s corrections to the typography, please increase the size to fit the full width of the document as it is too small to read.
Line 336: Please pluralise “feature” (“features”) and replace “is” with “are” to be grammatically correct
Conclusion section: I agree with the English language editor’s comments – this is not really a conclusion. A conclusion should sum up the key points in the context of existing findings and highlight implications for future research.
Comments on the Quality of English LanguageThe quality of English is ok - the English language editor has picked up on a lot of great points, but there are still a few issues surrounding word choice and interpretation, which make it difficult to read. Please see listed comments above.
Author Response
Reply to the reviewers
I would like to express my gratitude for your helpful comments and suggestions. We have carefully addressed and revised the previous manuscript. The revised parts are indicated in red letters.
Reviewer 1
Firstly, the title does not clearly articulate the content to the reader - what does "clinicopathological relevance of Epstein-Barr virus infections" mean? It has long been known that EBV is associated with NPC. What does this review add to the body of knowledge? What is unique or novel about this? Integrate THIS into your title.
→Thank you for pointing this out. In response to your suggestion, we have modified the title to read as follows: “Recent advances in the clinical implications of Epstein-Barr virus infection and their application to the diagnosis and treatment of nasopharyngeal carcinoma.”
Similarly with the abstract, this should draw out the key points addressed in the review, but it fails to do so clearly. I think it just needs to be more clearly stated.
→Thank you for your valuable comment. We have revised the abstract to better align with the content of the manuscript.
The subsections on LMP1 and BZLF1 are very short - too brief really. Please expand to include more recent, relevant references too.
→Your valuable comment is greatly appreciated. Since the two EBV genes are relevant to various aspects such as genomics, epigenomics, and immune-related subchapters, we initially provided a fundamental overview of their functions. However, we acknowledge that both LMP1 and BZLF1 sections are relatively short. Therefore, we have incorporated recent descriptions to enhance their coverage. Additionally, we have included Figure 1 to aid readers in comprehending the multifunctional LMP1-mediated signaling pathway activation and its relevance to the “hallmark of cancer.”
Then, there are numerous issues throughout, which are laid out line by line below:
In addition to the comments from the English language editor throughout:
Lines 21-23: This is not a complete sentence. (Or perhaps you mean “through” instead of “though”?)
→Thank you for bringing this to our attention. This phrase and the following sentence convey that “EBV is not typically produced in latently infected tumors. However, elevated concentrations of the anti-EBV antibodies and plasma EBV DNA have been used as biomarkers for EBV-associated NPC.” If necessary, we will make the suggested revision to the original sentence.
Line 32: Kaposi’s (not Kaposi)
→Thank you for addressing this issue. We have made the necessary correction.
Line 53: Formatting – please indent for consistency
→Thank you for addressing this issue. We have made the necessary correction.
Line 53: Rephrase as “EBV-associated WHOO II and III NPC are characterized…”
→Thank you for addressing this issue. We have made the necessary correction.
Line 66: Please add a reference at the end of this sentence (“…WHO II and III NPCs.”) – unless it is the same reference as at the end of line 69, in which case you can ignore this comment.
→Thank you for raising this point. The same reference [13] remains applicable.
Table 1: As the English language editor says, the Table needs a footnote with a list of abbreviations in alphabetical order. Please see author guidelines for more detail.
→Thank you for raising this point. We have added a list of abbreviations in Table 1.
Line 75: Please consider a different word than “adapted” as this sentence does not make sense.
→I appreciate your attention to this matter. We have replaced the word “adapted” with “incorporated,” which we believe provides better clarity.
Line 75: The following sentence is also inaccurate, “HPV is considered to contribute to WHO grad I NPC because EBV genomes are frequently missing from samples of this subtype.” The two facts are unrelated – you cannot possibly consider HPV causality merely because EBV genomes are missing. Please rephrase for clarity.
→Thank you for your important suggestion. We have deleted the sentence.
Line 83: This sentence lends nothing to the argument. Either expand it to give it more weight and meaning (e.g., WHY is NPC of interest?) or remove it.
→Thank you for your insightful suggestion. We have removed the sentence.
Line 88: Remove “the” before “latent”
→Thank you for your valuable comment. We have omitted “the.”
Line 88: Remove “The” before “EBV”
→Your meaningful comment is duly noted. We have omitted “the.”
Line 97: Your statement that any of the latent gene products influence tumorigenicity, but this is not quite true. EBNA1 does not promote tumorigenicity but rather episomal maintenance. Please be very clear and accurate in your writing.
→We value your perceptive comment. I acknowledge that the primary role of EBNA1 is the episomal maintenance of the EBV genome. However, some reports emphasize the relevance of EBNA1 to tumorigenic activity. Therefore, we have included “EBNA1” in the list of EBV gene products that influence tumorigenic properties. We have removed “EBNA1” as per your recommendation.
Line 110: Please rephrase the end of this sentence – which signaling? Presumably signaling by LMP1, but this needs to be made clearer.
→We are thankful for your valuable contribution. I meant the NF-κB signaling pathway. We have added “NF-κB.”
Line 114: The “k” in NF-kB is not in symbol format, whereas it is elsewhere. Please go through the entire document and ensure consistency throughout.
→Your insightful comment is recognized and appreciated. We have made the change to “NF-κB.”
Lines 114-115: This sentence is rather “lost” here on its own. There needs to be more context around this statement. What kind of somatic alterations? How does LMP1 overexpression drive cytotoxicity?
→We are grateful for your valuable remark. The reference 46 describe somatic mutations such as CYLD, TRAF3, NFKBI, and NLRC5, which activate NF-κB pathway, are detected in NPC tissue. And these mutations are mutually exclusive relationship with LMP1. We added these NF-κB activating somatic mutaions and we deleted the expression about the cytotoxic effect of LMP1 overexpression as it may not well established issue. We have made the necessary correction.
Section 4 & 5: Given the importance of LMP1 and BZLF1 on NPC pathogenesis, you have written very little about either gene product. I would recommend expanding these sections to be more comprehensive.
→Your constructive input is greatly appreciated. Since the two EBV genes are relevant to various aspects such as genomics, epigenomics, and immune-associated subchapters, we initially provided a fundamental overview of their functions. However, we acknowledge that both LMP1 and BZLF1 sections are relatively short. Therefore, we have added some recently reported descriptions.
Line 121: This is not 100% accurate – any of the lytic gene products influence NPC tumorigenicity – I would temper this statement. You made a similar claim earlier about latent gene products too (line 97), which also wasn’t accurate.
→We extend our gratitude for your valuable feedback. I agree that a few lytic gene products, such as BHRF1 and BRLF1, are suggested to influence tumorigenesis. I have revised the sentence accordingly.
Line 142: How can LMP1 and BZLF1 genes precede EBV infection in NPC? The virus is what encodes them, therefore the infection surely precedes their expression! Please rephrase
→We extend our gratitude for your valuable feedback. The revised sentence now reads, “LMP1 and BZLF1 are key EBV genes that drive EBV-infected epithelial cells toward NPC.”
Lines 142-143: In what way was the influence of these genes reversible? Do you have a reference for this?
→We thank you for your valuable observation. Transformation induced by the transient expression of oncogenes is generally temporary. We mentioned this in the sentence. We have revised the section to state, “However, the expression of EBV genes alone is insufficient for the development of clinical NPC. The accumulation of multiple irreversible somatic genetic alterations, in addition to EBV infection, is required for the development of NPC [22].
Line 203: Formatting – please indent for consistency
→We are grateful for your valuable remark. We have made the necessary correction.
Line 231: Please reconsider use of the word “should” – this sentence could be more concise (e.g., “Malignant tumor cells begin constructing the TIME once they are recognized as immune cell targets.”)
→We appreciate your insightful input. We have made the necessary correction.
Line 243: Repetition of derivatives of the word “aggregate” – consider rephrasing
→Thank you for your valuable comment. We have revised the sentence to read “Upon activation, LMP1 localizes and aggregates within lipid rafts on the cell membrane.”
Line 258: What is meant by “leaky EBV gene expression”?
→Thank you for pointing this out. The term “leaky” describes gene products being produced to some degree when their transcription and translation are silenced. I have changed “leaky” to “some.”
Lines 258-259: Please incorporate “This is one theory” into the preceding sentence (e.g., “One theory states that…”)
→Your valuable comment is greatly appreciated. We have incorporated the suggestions made by the reviewer.
Line 262: Please add “gene” after “env”
→Thank you for bringing this to our attention. We have added “gene” after “ebv.”
Line 262: Please remove “the” before “latently”
→Thank you for addressing this issue. We have made the necessary correction.
Line 263: Why separate out LMP1 and LMP2A if they both transactivate the same gene? If this is in fact correct, please combine (e.g., “both LMP1 and LMP2A are able to transactivate…”
→Thank you for addressing this issue. We have made the revisions as suggested by the reviewer.
Line 264: LMP1 and LMP2A are expressed in NPC cells – please specify which subtype
→Thank you for raising this point. We have revised it to “Both LMP1 and LMP2A are expressed in NPC cells.”
Lines 271-272: Please add appropriate references for these statements, unless it is incorporated within ref 110, in which case ignore this comment
→I appreciate your attention to this matter. Yes, these statements are described in reference 110.
Line 290: Please find an alternative word to “acceptable” as this does not make sense
→Thank you for your important suggestion. We have changed “acceptable” to “improved.”
Line 313: Please remove “as mentioned in the previous chapter” as this is not relevant here (I’m assuming this came from a student thesis?!)
→Thank you for your insightful suggestion. We mentioned this in chapter 2. However, as the reviewer suggested, we have removed the phrase.
Table 2: In addition to the English language editor’s corrections to the typography, please increase the size to fit the full width of the document as it is too small to read.
→Your meaningful comment is duly noted. We have corrected the typo, and increased the size of Table 2.
Line 336: Please pluralise “feature” (“features”) and replace “is” with “are” to be grammatically correct
→Your meaningful comment is duly noted. We have made the necessary correction.
Conclusion section: I agree with the English language editor’s comments – this is not really a conclusion. A conclusion should sum up the key points in the context of existing findings and highlight implications for future research.
→We value your perceptive comment. We have made the revisions as suggested by the reviewer and added key points to this review.
Comments on the Quality of English Language
The quality of English is ok - the English language editor has picked up on a lot of great points, but there are still a few issues surrounding word choice and interpretation, which make it difficult to read. Please see listed comments above.
→ We sincerely appreciate your valuable comments and suggestions.

Reviewer 2 Report
Comments and Suggestions for Authors
The manuscript entitled "Nasopharyngeal carcinoma: Recent understanding of the clinicopathological relevance of Epstein-Barr virus infections" by Yoshizaki et al is a review article related to EBV infection. The review is well written and well presented, however it needs a minor revision. My suggestions are:
1. The progress of drugs development for the prevention and cure of EPV infection should be discussed in detail in separate section.
2. Future recommendations should be addressed.
3. Conclusion is very precise, it must be expanded.
3. In the text, the diagrams and flow charts are missing, For instance, in sections 3, 4, and 5, the information in the text should be presented in diagrams as well to make it interesting for the readers.
Author Response
Reply to the reviewers
I would like to express my gratitude for your helpful comments and suggestions. We have carefully addressed and revised the previous manuscript. The revised parts are indicated in red letters.
Reviewer2
- The progress of drugs development for the prevention and cure of EPV infection should be discussed in detail in separate section.
→We are thankful for your valuable contribution. We have added a new subchapter that describes the progress of vaccine development against EBV infection in chapter 6. However, there has been no significant progress in the development of anti-EBV drugs, so we have only provided a brief mention of it in subchapter 4.
- Future recommendations should be addressed.
→Your insightful comment is recognized and appreciated. We have expanded the “Conclusion” section to include future recommendations.
- Conclusion is very precise, it must be expanded.
→Your constructive input is greatly appreciated. We have made the necessary revisions as recommended by the reviewer.
- In the text, the diagrams and flow charts are missing, For instance, in sections 3, 4, and 5, the information in the text should be presented in diagrams as well to make it interesting for the readers.
→We extend our gratitude for your valuable feedback. We have integrated chapter 3, 4, 5, and 6, and expanded the content. Additionally, we have included Figure 1, which will assist readers in understanding the multifunctional LMP1-mediated signal pathway activation and its relevance to the “hallmarks of cancer.”
。

Reviewer 3 Report
Comments and Suggestions for Authors
The manuscript by Yoshizaki et al., reviews the role of Epstein-Barr virus (EBV) infection in nasopharyngeal carcinoma addressing the histopathology, immunology, prognosis, and diagnostic features of nasopharyngeal carcinoma associated with EBV infection.
I agree and align with the comments provided in the manuscript (probably by another reviewer). Importantly: the conclusion section needs to be rewritten summarizing the findings noted in the review and the gene names need to be italicized throughout.
Comments on the Quality of English Languageminor grammatical corrections are needed throughout the manuscript
Author Response
Reply to the reviewers
I would like to express my gratitude for your helpful comments and suggestions. We have carefully addressed and revised the previous manuscript. The revised parts are indicated in red letters.
Reviewer 3
We are thankful for your valuable comments. We have revised according to the reviewers recommendation.
Round 2
Reviewer 1 Report
Comments and Suggestions for Authors
Thank you for making the revisions outlined in my previous review of your manuscript. This is a great improvement, and we are happy to accept it for publication now.
Comments on the Quality of English LanguageYou have made significant improvements to the written English, many thanks.